Microbiology **Spectrum**

COMMENTARY
# The Challenge to Control Emergence of Antibiotic Resistance in Virulent *Escherichia coli* Isolates in Latin America

Alfredo G. Torres[a]

[a]University of Texas Medical Branch, Department of Microbiology and Immunology, Galveston, Texas, USA

**ABSTRACT** The Latin American Coalition for *Escherichia coli* Research (LACER) was created as a network of investigators using One Health approaches trying to understand infections caused by regional *E. coli* isolates and to sound the alarm due to the evolution of strains that are multiresistant to antibiotics (resistome) that also display different virulence profiles (virulome). After the COVID19 pandemic, a major concern by investigators has been the appearance of more virulent and resistant strains. Recently, a paper published in Microbiology Spectrum by Brazilian investigators (Fuga B., et al. Microbiol Spectr 10: e0125621, 2022, https://doi.org/10.1128/spectrum.01256-21) has used a genomic approach to demonstrate that during a period of 45 years, a wide resistome and virulome has converged, resulting in the appearance and persistence of high-risk clones affecting humans, animals and the environment, and its rapid dissemination is becoming an unattended international threat.

**KEYWORDS** *Escherichia coli*, Latin America, antibiotic resistance, pathogenic, resistome, virulome

It is well documented that antibiotic resistance is one of the largest threats to human and animal health, impacting the environment and causing significant challenges, including prevention and treatment of the infections caused by these pathogens. The Latin American region faces a significant challenge with high levels of antimicrobial resistance among important Gram-negative organisms, including *E. coli* strains. In the past 50 years, antibiotic resistant *E. coli* strains have increased and multidrug-resistant *E. coli* has spread across the American continent (1, 2). However, the association of these resistance mechanisms (resistome) with the capacity of the strains to cause infections and disease (virulome) has been neglected. Therefore, coordinated efforts from the scientific community are needed to understand the different evolving isolates, but also to establish effective strategies to diagnose, treat and combat the dissemination of strains that can cause regional or international outbreaks.

## EVOLUTION OF THE PATHOGENIC *E. COLI* STRAINS AND THE LATIN AMERICAN COALITION FOR *ESCHERICHIA COLI* RESEARCH (LACER)

Initial studies in the mid-1940s led to discussions about the role of *E. coli* as a pathogen in Mexico and the rest of Latin America, particularly those impacting the intestinal health of humans and animals (3). With the advances of serology and molecular methods, it was demonstrated that specific serogroups and serotypes of *E. coli* were responsible for a variety of signs and symptoms, from diarrhea to urinary tract infections and from meningitis to other systemic infections. The emergence and evolution of several pathogenic *E. coli* strains associated with animal and human infections demonstrated that: (i) groups of related pathogenic *E. coli* (pathogroups) strains were responsible for most infections caused by these bacterial species; (ii) diverse virulence phenotypes expressed during infection were characteristics and defined each one of these pathogroups; (iii) the geographical distribution of pathogroups existed in Latin America and evolution of new isolates was defined by the dominant

Address correspondence to altorres@utmb.edu.

The authors declare no conflict of interest.

For the article discussed, see https://doi.org/10.1128/Spectrum.01256-21.

*The views expressed in this Commentary do not necessarily reflect the views of the journal or of ASM.*

pathogroup and presence of distinct virulence traits; (iv) acquisition of mobile elements or accumulation of point mutations accelerate the development of antibiotic resistance of some of these strains (1, 2). Due to the limited resources in Latin America to perform investigation and implement public health programs designed for understanding and controlling the dissemination of these pathogenic *E. coli* strains, the investigators in the region realized that collaboration was the only way to tackle this public health problem (4). As a result, the Latin American scientists responded to the threat of continuously evolving pathogenic *E. coli* strains by establishing in 2009 the Latin American Coalition for *Escherichia coli* Research (LACER), a multidisciplinary group of approximately 70 research groups in 11 Latin American countries and the USA, applying principles of One Health (5, 6). LACER is integrating animal, human and environmental health for the mutual benefit of all individuals, while reducing the impact of human and animal diseases. As a result of these collaborations, LACER investigators have monitored and sounded the alarm about the emergence of new *E. coli* strains with dynamic genomes that are able to transfer or acquire important antibiotic resistance or virulence factors (1, 2). In recent years, several Latin American groups have evaluated the appearance of these emergent strains, their relationship to natural reservoirs and the diversity of their virulomes and resistomes, as the recent paper published in Microbiology Spectrum (7).

## GENOMIC SURVEILLANCE OF BRAZILIAN *E. COLI* ISOLATES DEMONSTRATED EMERGENCE AND DISSEMINATION OF HIGH-RISK CLONES

Few Latin American countries have contributed in such a significant way to understand about *E. coli* pathogenesis like Brazil. From the initial seminal paper describing the patterns of adherence that different pathogenic *E. coli* strains displayed on tissue cultured cells (8) to the latest paper (7), presenting epidemiological and clinical evidence of the convergence of virulome and resistome traits contributing to persistence and dissemination of high-risk *E. coli* isolates, Brazilian investigators have always been at the forefront of investigation of this pathogen (9). The manuscript by Fuga, B., et al. (7), utilized principles of the One Health initiative to interrogate the genomes of a historical collection of Brazilian *E. coli* strains expanding 45 years. In such analysis, the investigators focused their efforts defining the virulence traits (virulome), the presence of antibiotic resistance (resistome) and mobile elements (mobilome), as well as defining the strains' sequence types and evolutionary properties (phylogenomics). The genomic information provided by 167 strains obtained from human, animals and the environment, circulating in 40 Brazilian cities, and 16 highly populated states, depicted a very problematic picture: (i) antibiotic resistance is broadly distributed, (ii) strains had acquired specific combinations of virulence and resistance genes, (iii) the conditions in the region are adequate for the expansion of high-risk clones, and (iv) phylogenomics demonstrated human and nonhuman samples clustering together and suggestive of a zoonotic spillover.

The elegant study by Fuga, B., et al., is warning the research community as well as the public health systems, that in Brazil high-risk clones had spread containing a broad antimicrobial resistance, including several endemic $\beta$-lactamases (e.g., $bla_{CTX-M}$ ESBL) and colicin (*mcr-1*) genes, and the rapid and troublesome expansion of carbapenemase ($bla_{KPC-2}$ and $bla_{NDM-1}$) genes. Further, there is a convergence of highly virulent *E. coli* strains with resistome genes of concern, making them a significant health treat, because spillover of these strains into the environment will create the conditions for the emergence of more virulent strains with zoonotic potential. International experts have been concerned that the COVID19 pandemic could undo much of the nation's progress on antibiotic resistance, due to the difficulty monitoring antibiotic-resistant foodborne bacteria. The current paper demonstrated that at least in Brazil, the antimicrobial resistance problem has worsened.

## CLOSING REMARKS

This comprehensive manuscript is a wake-up call not just for Latin America but the world, reminding us that pathogenic *E. coli* are constantly evolving as they exchange genetic material in the environment and as a result, new emerging isolates are able to cause outbreaks and can emerge in any place and at any time. If these evolving new *E. coli* clones acquired the right combination of virulence factors and antibiotic resistant markers, the result can be lethal

for the human and animals host and for the environment. We need to work as a unified scientific community to stop this threat.

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
