## [Reviewer comments · Microbiology Spectrum]

Microbiology Spectrum

The challenge to control emergence of antibiotic resistance in virulent *Escherichia coli* isolates in Latin America

Alfredo Torres

Corresponding Author(s): Alfredo Torres, UTMB

Review Timeline:

Submission Date:	April 25, 2022
Editorial Decision:	May 23, 2022
Revision Received:	May 24, 2022
Accepted:	June 14, 2022

Editor: Christina Cuomo

Reviewer(s): The reviewers have opted to remain anonymous.

Transaction Report:

DOI: <https://doi.org/10.1128/spectrum.01506-22>

May 17, 2022

Dr. Alfredo G Torres
UTMB
Microbiology and Immunology
301 University Blvd
Galveston, Tx 77555-1070

Re: Spectrum01506-22 (The challenge to control emergence of antibiotic resistance in virulent *Escherichia coli* isolates in Latin America)

Dear Dr. Alfredo G Torres:

Thank you for submitting your commentary to Microbiology Spectrum. There a few minor comments with suggested changes to wording (see below) that I would appreciate you addressing in a revision.

Please use this link to submit your revised manuscript - we strongly recommend that you submit your paper within the next 60 days or reach out to me. Detailed instructions on submitting your revised paper are below.

Link Not Available

Sincerely,

Christina Cuomo

Journals Department
Reviewer comments:

Reviewer #1 (Comments for the Author):

This is a clear and engaging commentary on the combined threat of antibiotic resistant and virulence E coli isolates in Latin America.

Minor comments:

Line 36: Change "strains multi-resistant" to strains that are multi-resistant or strains displaying multi-resistance

Line 38: change "big" to major

Line 39; Change "Now" to Recently

Line 41: The phrase "has converged to the appearance" is unclear. Perhaps this could be reworded as have converged, resulting in the appearance

Line 79: Change "accelerate the antibiotic resistance" to accelerate the development of antibiotic resistance

Line 80: Change "public health programs understanding" to public health programs designed for understanding

Line 83: Change "thread" to threat

Line 88: Change "this collaborations" to these collaborations

Line 89: Change "sound the alarm" to sounded the alarm

Line 103-4: Change "the Brazilian investigators" to Brazilian investigators

Staff Comments:

Preparing Revision Guidelines

Please return the manuscript within 60 days; if you cannot complete the modification within this time period, please contact me. If you do not wish to modify the manuscript and prefer to submit it to another journal, please notify me of your decision immediately so that the manuscript may be formally withdrawn from consideration by Microbiology Spectrum.

Response to reviewers of commentary Spectrum01506-22:

Thanks for reading our commentary. We have modified accordingly all the minor comments listed below in the revised manuscript.

Minor comments:

Line 36: Change "strains multi-resistant" to strains that are multi-resistant or strains displaying multi-resistance

Line 38: change "big" to major

Line 39; Change "Now" to Recently

Line 41: The phrase "has converged to the appearance" is unclear. Perhaps this could be reworded as have converged, resulting in the appearance

Line 79: Change "accelerate the antibiotic resistance" to accelerate the development of antibiotic resistance

Line 80: Change "public health programs understanding" to public health programs designed for understanding

Line 83: Change "thread" to threat

Line 88: Change "this collaborations" to these collaborations

Line 89: Change "sound the alarm" to sounded the alarm

Line 103-4: Change "the Brazilian investigators" to Brazilian investigators

June 14, 2022

Dr. Alfredo G Torres
UTMB
Microbiology and Immunology
301 University Blvd
Galveston, Tx 77555-1070

Re: Spectrum01506-22R1 (The challenge to control emergence of antibiotic resistance in virulent *Escherichia coli* isolates in Latin America)

Dear Dr. Alfredo G Torres:

Your manuscript has been accepted, and I am forwarding it to the ASM Journals Department for publication. You will be notified when your proofs are ready to be viewed. Thank you for submitting this and for your patience while we worked out setting up the Commentary format for Spectrum.

Sincerely,

Christina Cuomo
Editor, Microbiology Spectrum
